# BlihIA—A Novel Type I Restriction-Modification System from *Bacillus licheniformis* Is Sensitive to In Vitro Inhibition by ArdB Antirestriction Protein

**DOI:** 10.3390/ijms26178674

**Published:** 2025-09-05

**Authors:** Anna Kudryavtseva, Rodion Berezov, Anna Utkina, Oksana Kotovskaya, Mikhail Skutel, Anna Trofimova, Artem Isaev, Ilya Manukhov

**Affiliations:** 1Moscow Center for Advanced Studies, Moscow 121059, Russia; berezov.rodion.2001@gmail.com (R.B.); manukhovi@mail.ru (I.M.); 2Center for Molecular and Cellular Biology, Moscow 121205, Russia; ktv.xna@gmail.com (O.K.); mskutel@yandex.ru (M.S.); annat@genebiology.ru (A.T.)

**Keywords:** restriction-modification, BlihIA, ArdB, antirestriction proteins, ONT sequencing, methylation

## Abstract

Type I restriction-modification (RMI) systems play a crucial role in bacterial defense against mobile elements by distinguishing self and foreign DNA through sequence-specific methylation and cleavage. Here, we characterize BlihIA, a novel RMI system from *Bacillus licheniformis* DSM13 which features redundancy in its *hsdS* gene copies. Using ONT sequencing, we identify the bipartite recognition site of BlihIA as RTAC(N)_5_GCT. We demonstrate the system’s activity both in vivo through efficiency of plaquing (EOP) assay and in vitro in a nuclease reaction with purified BlihIA complex. Notably, mutation of the recognition site abolished in vitro DNA cleavage, confirming sequence specificity. Furthermore, we show that the antirestriction protein ArdB from plasmid R64 effectively prevents DNA cleavage by BlihIA, suggesting a direct mechanism of inhibition. This study provides the first functional characterization of a novel RM system BlihIA, extending the diversity of RM systems in *Bacillus* species and suggesting potential applications for improving genetic transformation in industrial strains.

## 1. Introduction

Restriction-modification (RM) systems play a crucial role in protecting prokaryotes from the invasion of mobile genetic elements (bacteriophages, conjugative plasmids, and transposons) [1]. These systems are usually composed of two modules: restriction endonuclease subunit introduces double-stranded breaks in non-modified DNA, while the modification component provides protection for the host DNA [2]. RM systems could represent the part of a core genome or could be mobile, since they are often encoded by plasmids and prophages [3]. Depending on the structure of restriction and modification complexes, recognition site characteristics, and cleavage position, restriction-modification systems are divided into four types and multiple subtypes [4].

Type I restriction-modification (RMI) systems represent an important barrier to horizontal gene transfer and are encoded by at least 50% of bacteria [5,6]. RMI systems consist of three subunits: HsdS (the specificity subunit, which recognizes target DNA via two target recognition domains (TRDs)); HsdM (methyltransferase subunit); and HsdR (the restriction subunit, combining nuclease and translocase activities). Together, HsdSM_2_ subunits comprise a methylation complex that can interact with one or two HsdR subunits to assemble a restriction complex [2,3].

Binding to a non-methylated recognition site initiates ATP-dependent DNA translocation by a restriction complex. Upon collision with another restriction complex, or blockade of further translocation due to topological reasons, each HsdR subunit introduces a DNA nick at the collision site, resulting in dsDNA break in between two recognition sites, sometimes a few kilobases (kb) away from the initial complex binding site [2]. Circular plasmids bearing only one recognition site also can be cleaved; once the entire plasmid is pulled through the RMI restriction complex, further translocation becomes impossible, resulting in cleavage at the stopping point [2]. The RMI recognition motif is typically non-palindromic and consists of two specific 3–5 nucleotide-long regions, each recognized by individual Target Recognition Domains (TRDs) of the HsdS subunit, interspaced by a gap of 6–8 random nucleotides [2]. About 10% of RMI loci contain repeated sequences potentially prone to phase variation, and redundancy in *hsdS* gene copies has been found for ~4% of RM I operons, allowing these systems to modulate their site specificity [7,8].

Mobile genetic elements evolved multiple strategies to counter restriction by RMI systems: from avoidance of recognition sites to expression of their own methyltransferases or deliberate antirestriction proteins [9,10]. For example, DNA mimic ArdA and Ocr proteins are known to bind HsdMS complexes and inhibit restriction both in vitro and in vivo [11,12]. In contrast, a broad group of plasmid-encoded antirestriction proteins from the ArdB/KlcA family was reported to be active only in vivo [13]. The antirestriction mechanism of ArdC, a protein containing ssDNA binding and metalloprotease domain, is currently unknown [14], while the antirestriction effect of an “*ardD*” gene was recently explained through activation of restriction alleviation or programmed proteolysis of restriction complexes [15,16].

Despite RM systems’ abundance and important role in controlling horizontal gene transfer, phage resistance, and even gene expression, their activity and specificity in many industrially relevant bacteria remain understudied. *Bacillus licheniformis* is a bacterium of significant industrial relevance that is often used as a workhorse in the production of enzymes such as subtilisin and amylases [17]. *B. licheniformis* DSM13 strain encodes at least two predicted clusters of RMI systems with the following coordinates: 749.489–756.576 (named BlihIA) and 4.163.454–4.171.190 (named BlihIB). BlihIA system is particularly interesting due to the presence of an additional *hsdS* gene copy (Figure 1). While HsdSIA subunit is complete and contains both TRDs, HsdSIB subunit comprises just a half of the usual HsdS protein, bearing one TRD (Figure 1).

In a recent study, we demonstrated that both BlihIA and BlihIB systems are active upon expression in *E. coli* [18], and investigated their sensitivity to antirestriction proteins [18]. The aim of the present work is to identify and confirm the BlihIA recognition site and to confirm whether ArdB inhibits BlihIA DNA cleavage in vitro.

Given that many Gram-positive bacteria, including those of the genus *Bacillus*, are used as synthetic biology platforms, and their transformation is often challenging [19], these results could be used to design plasmids that overcome BlihIA resistance via restriction sites avoidance or via expression of antirestriction proteins.

## 2. Results

### 2.1. BlihIA Restriction Activity In Vivo

*B. licheniformis* DSM13 strain contains at least two predicted clusters of RMI genes: BlihIA and BlihIB. In this study, we focused on the BlihIA system due to the intriguing presence of an additional, incomplete *hsdSIA* gene copy encoding only one TRD (Figure 1).

After the full BlihIA cassette was cloned into an expression vector with the C-terminal 6xHis tag attached to the HsdR subunit, we confirmed the activity of the system in a phage λ efficiency of plaquing (EOP) assay (Table 1). The results, verified by an ANOVA *t*-test, demonstrate that 6xHis tag as well as the change in the expression vector does not affect the restriction activity of BlihIA against phage λ_0_ compared to the previously described system pIRal-2_RM-Ia [18]. Thus, the pIR-DPAl-BlihIA-His-TAG construct is suitable for further in vitro studies.

### 2.2. Identification of BlihIA Methylation Site

To identify the BlihIA methylation site, we sequenced genomic DNA of *E. coli* TG1 expressing the full BlihIA system or DNA from a control culture carrying an empty vector. The DNA was sequenced using an Oxford Nanopore (ONT) platform that allows direct identification of modified bases. To accurately map sequencing reads, we assembled the genome of laboratory TG1 strain and noticed that it differs from a deposited refseq sequence (NZ_CP128218.1) (Figure 2). In addition to multiple single-nucleotide polymorphisms (Appendix A), our strain demonstrated different positions of one copy of the cut-and-paste IS1 element (located within the *fhuA* gene in the reference strain, and inside the *dcyD* gene in our laboratory strain), and the presence of the 31-kB region encoding Type I fimbria synthesis loci, which was absent in the reference TG1 genome. Inactivation of the flagellar genes and IS1 transposition events are often observed in the evolving populations of *E.coli* [20].

RMI systems could be subjected to phase variation, changing the specificity of the HsdS subunit [7], and since ONT sequencing allows direct identification of genomic rearrangements, we applied the PhaVa tool v.0.2.3 [21] to investigate whether two *hsdS* genes within the BlihIA locus also could be subjected to programmed phase variation in an *E.coli* host. This analysis identified events of promoter inversions within the Type I fimbriae loci, in agreement with a previous observation [22], but no genomic rearrangements were associated with the plasmid-encoded BlihIA locus.

Next, we used the assembled TG1 genome to map methylation and identified the expected motifs associated with m5C Dcm CCWGG sites and m6A Dam GATC sites, as well as a novel m6A motif present only in the BlihIA, but not a control empty vector culture (Figure 3). This bipartite motif interspaced with five random nucleotides, RT**A**C(N)_5_GC**T**, carries methylation at both DNA strands (at the underlined positions) and is typical for the RMI systems. Therefore, we assigned this motif to the variant of BlihIA system carrying both HsdS subunits.

### 2.3. BlihIA Restriction Activity In Vitro

Identification of BlihIA recognition site allowed us to proceed with in vitro investigation of restriction activity using a DNA substrate containing an HsdS recognition site. We purified the BlihIA complex using His-tag affinity followed by ion-exchange chromatography. Figure 4A represents the results of BlihIA purification and the migration patterns of purified proteins suggest co-purification of both HsdSIA and HsdSIB subunits. To validate this unexpected observation, we extracted protein bands from the gel and analyzed them via MALDI-TOF mass spectrometry. This confirmed the identity of all BlihIA subunits and revealed the presence of Dps, a highly abundant nucleoid-binding protein, which was likely nonspecifically bound to the resin.

Although the function of the HsdSIA subunit remains unknown, it should be co-expressed with HsdSIB and could participate in the target recognition process. Purified BlihIA complex was mixed with the plasmid pIR-DPAl_Blih_site, bearing a single recognition site RTACNNNNNGCT. The results of an enzymatic restriction are presented in Figure 4B and demonstrate gradual accumulation of restricted linear plasmid. We further created a pIR-DPAl_Blih_Mut plasmid that contained a point mutation in the BlihIA recognition site (TACNNNNN**T**CT). As can be seen in Figure 4C, this mutation completely eliminated the in vitro restriction activity.

### 2.4. ArdB Protein Inhibits BlihIA Restriction Activity In Vitro

In a previous work [18], we demonstrated that antirestriction protein ArdB from conjugative plasmid R64 (ArdB (R64)) inhibits anti-viral immunity of the BlihIA system in vivo. Here, we decided to investigate this phenomenon in vitro (Figure 5). While previous studies failed to demonstrate in vitro antirestriction activity of the proteins from the ArdB/KlcA family [13], we were able to observe complete inhibition of BlihIA in the presence of 30× molar excess of ArdB (Figure 5A). To validate this surprising observation, we used a mutant version of ArdB (R64) protein that lacks the highly conserved C-terminal D141 residue required for in vivo antirestriction activity [25]. BlihIA restriction reaction in the presence of ArdBΔD141 proceeded efficiently, confirming a specific inhibitory effect of the ArdB (R64) protein (Figure 5B).

## 3. Discussion

Bacteria of the genus *Bacillus* are among the most important industrial microorganisms, and are widely used in enzyme and metabolite production, agriculture, and other applications [26,27]. In addition to *B. subtilis*, other species such as *B. licheniformis* [28], *B. megaterium* [29] and *B. pumilus* [30] are widely used as protein-producing organisms for industrial purposes. The efficiency of transformation of bacilli with foreign DNA is influenced by several factors: cell wall structure [31], activity of natural competence genes *comK* and *comS* [32], as well as restriction-modification or other immunity systems. It was shown that mutations in *hsd* genes in *B. licheniformis* MS1 improve transformation efficiency [33]. RM systems protect bacterial cells against phage infection, yet they also broadly restrict acquisition of external DNA-bearing recognition sites, and thus limit horizontal gene transfer.

Our study characterizes the specificity and in vitro activity of a novel RMI system BlihIA from *B. licheniformis*, and these results can be used for improving the transformation to *Bacillus* hosts. First, using ONT sequencing, we identified the BlihIA system recognition motif (RTACNNNNNGCT), and determined that avoidance of such target sites in the plasmid could enhance transformation of *B. licheniformis* DSM13 host. Second, we demonstrated that BlihIA restriction could be inhibited by ArdB antirestriction proteins both in vivo and in vitro. This result is especially interesting, since previous studies failed to demonstrate in vitro antirestriction activity of the proteins from ArdB/KlcA family, leading to hypotheses that some additional components present in bacterial cell are required to support the activity [13]. ArdB’s antirestriction mechanism is still undetermined, but it was previously shown that ArdB, and not ArdBΔD141 lacking antirestriction activity, could nonspecifically interact with DNA [34]. For other antirestriction proteins, like DNA-mimic Ocr, it was shown that co-transformation with sensitive plasmid could inhibit resident RMI systems in native hosts, significantly improving transformation efficiency [35]. Our results suppose that ArdB proteins could also be used for this purpose.

Finally, we identified that in addition to the complete HsdSIB subunit, BlihIA system encodes a second copy of the *hsdS* gene, encoding the partial HsdSIA subunit. While HsdS subunits usually encompass two target recognition domains, HsdSIA encodes only one TRD. A truncated HsdS subunit called S_1/2_ was reported as a fusion to the HsdM protein in RMI operon from *Vibrio vulnificus* YJ016 [36]. Assembly of two HsdM_1_HsdS_1/2_ fusion proteins in one complex recapitulates a classical HsdM_2_HsdS_1_ assembly. Truncated single-TRD HsdS subunits can self-dimerize, resulting in the assembly of a complete HsdS protein with identical TRD, recognizing palindromic motifs [37], suggesting that the presence of an additional copy of truncated HsdS could extend the sequence specificity of the BlihIA system. However, models explaining how two different types of HsdS subunits interact within one RM systems are currently lacking. From the present analysis, it is impossible to tell if both full-length and truncated HsdS subunits are present in the BlihIA restriction complex at the same time or if two alternative types of complexes exist (Figure 4A). A similar operonic organization of the RMI locus encoding full-length and truncated HsdS was recently reported in *Staphylococcus xylosus* and both subunits were shown to be necessary for methylation [38]. In addition, HsdS subunits encoding three TRDs were reported in the literature [7], suggesting the existence of non-canonical assemblies of RMI restriction complexes. It could be speculated that truncated subunit could play a regulatory role by forming a hybrid dimer with one of the TRDs of the full-length HsdS. Whether such hybrid HsdS dimers exist and if they play a regulatory or sequence specificity switching role in the BlihIA system will be the focus of further study.

## 4. Materials and Methods

### 4.1. Bacterial Strains and Plasmids

For cloning and subsequent expression of target genes, the following bacterial strains were used: *E. coli* TG1 (*upE hsdD5 thi* Δ (*lac-proAB*) *[F′ traD36 proAB* + *lacIqZ*Δ*M15]*) and *E. coli* NiCO21(DE3) (*can::CBD fhuA2 [lon] ompT gal* (*λ DE3*) *[dcm] arnA::CBD slyD::CBD glmS6Ala* ∆*hsdS λ DE3* = *λ sBamHIo* ∆*EcoRI-B int::(lacI::PlacUV5::T7 gene1) i21* ∆*nin5*). The plasmids used in this study are listed in Table 2.

### 4.2. The Cloning of the BlihIA Genes

The *blihIA* genes were cloned from the plasmid pIRal-2_RM-Ia (Table 2) using the thermo-inducible pIR-DPAl vector. As a result, the construct pIR-DPAl-BlihIA-His-TAG was obtained, where the expression of *blihIA* genes is controlled by a thermo-inducible promoter and a 6xHis tag is added to the C-terminal end of the BlihIA HsdR subunit.

### 4.3. BlihIA Restriction Activity In Vivo

The restriction activity of the newly cloned BlihIA RMI system in the pIR-DPAl vector was measured using the efficiency of plaquing (EOP) assay with unmodified phage λ_0_, as described previously [18]. The bacteriophage λ_vir_ was a kind gift from Prof. R. Devoret (France). Unmodified phage λ_0_ was grown on *E. coli* TG1. The restriction values were estimated by comparing λ_0_ bacteriophage titers in *E. coli* TG1 cells carrying no plasmids with TG1 cells containing plasmids expressing the BlihIA system. The “double-overlay agar layer” method was used to estimate phage infectivity [40].

### 4.4. ONT Sequencing and Data Analysis

To identify the DNA specificity of the BlihIA system, we performed sequencing on the Oxford Nanopore (ONT) platform, followed by the detection of methylation motifs. Genomic DNA of *E. coli* TG1 carrying a control empty vector or pIR-DPAl-BlihIA-His-TAG was purified from 2 mL of overnight cultures, as described previously [41]. Total DNA libraries were prepared from DNA treated with XbaI (Thermo Scientific, Waltham, MA, USA) using the Native Barcoding Kit 24 V14 (SQK-NBD114-24) with enrichment of long fragments using long fragment buffer according to the manufacturer’s instructions. The DNA library was sequenced using R10.4.1 flow cell (FLO-MIN114) on the MinION device with MinKNOW v23.11.2.

Raw ONT sequencing signals were basecalled with Dorado v.0.8.3 [https://github.com/nanoporetech/dorado (accessed on 4 June 2025)], using an accurate basecalling model (sup, v.5.0.0) and modification detection models for m6A, 4mC, and 5mC (v.2).

Basecalled long reads were trimmed with Porechop v.0.2.4 [https://github.com/rrwick/Porechop (accessed on 4 June 2025)] and filtered with Filtlong v0.2.1 [https://github.com/rrwick/Filtlong (accessed on 4 June 2025)].

Before modification analysis, we assembled the TG1 genome with obtained long reads via Unicycler v.0.5.0 [42], and the assembly was polished with NextPolish v.1.4.1 [43], resulting in one contig chromosome with a size of 4.522.373 bp (~128 coverage) and an external plasmid vector with a size of 10.602 bp. The completeness of the assembly was examined with gVolante Web-service [44] with OrthoDB v.10 [45] dataset of *Enterobacterales* (obtained scores in BUSCO format are C:99.6%[S:99.1%,D:0.5%],F:0.5%,M:0.1%). Completeness and contamination were also assessed with CheckM v.1.2.3 (*E. coli* as a marker lineage) [46], showing 98.31% completeness and 0.15% contamination. Then, the assembly was annotated with Bakta v.1.9.4 [47]. A search of possible DNA inversions was performed with PhaVa v.0.2.3 [21]. The similarity between obtained assembly and deposited refseq assembly (NCBI Assembly ID: GCF_030316195.1) was assessed via alignment with Minimap2 v.2.28-r1209 [48], and further annotation of variants with SnpEff v5.0e [49]. Visualization of variants was performed with pyCirclize v.1.9.1 [https://github.com/moshi4/pyCirclize (accessed on 4 June 2025)] as a Python v.3.11 package and visualization of big insertions and deletions was performed with gggenomes v.1.0.0 [50] library in R v4.2.3 [https://www.R-project.org/ (accessed on 4 June 2025)].

### 4.5. Analysis of Methylation Patterns

Basecalled reads were analyzed for methylation according to MicrobeMod v.1.0.3 protocol [23]. Briefly, reads were extracted with samtools fastq (SAMtools v.1.20, [51]) and mapped on the reference genome assembled from ONT data with Minimap2 (Minimap2 v.2.28-r1209, [48]. Obtained alignments were sorted with samtools and used as an input for MicrobeMod call_methylation workflow with the following parameters: a minimal strand coverage of 10, ModKit methylation confidence of 0.66, a methylation percentage cut-off of 0.66, and a STREME motif detection cut-off of 0.7 (STREME version 5.5.5, [52]). Later analysis of site modification fraction, as a fraction of methylated reads covering one particular position of a reference genome, was performed with in-house python3 script based on ModKit output (ModKit v.0.3.1, [https://github.com/nanoporetech/modkit (accessed on 4 June 2025)]), obtained as a part of MicrobeMod workflow. Positions of the predicted motifs were found with the SeqKit locate tool (SeqKit v.2.10.0 [53]) and intersected with methylation positions with bioframe v.0.7.2 [54]. Distributions of reads were visualized with Matplotlib v.3.8.0 [55].

### 4.6. BlihIA Complex Purification

*E. coli* NiCO21(DE3) pIR-DPAl-BlihIA-His-TAG cells were grown in 500 mL of LB medium containing 10 μg/mL kanamycin at 37 °C for 5 h. After that, BlihIA expression was induced by lowering the temperature to 22 °C, followed by further growth for 22 h.

After incubation, cells were pelleted by centrifugation at 8000× *g* for 10 min. The average mass of the cell pellet per flask was 2.2 g.

The harvested cells were resuspended in a ratio of 1:10 *w*/*v* in 100 mM Tris-HCl buffer pH 8.0 with 10% glycerol, 10 mM imidazole, and 1 mM PMSF. Cells were then lysed using an M110P Microfluidizer Processor (Microfluidics, Westwood, MA, USA). The resulting lysate was centrifuged at 30,000× *g* at 4 °C for 10 min.

The BlihIA complex purification was performed using Ni-NTA agarose (elution buffer: 100 mM Tris-HCl buffer pH 8.0, 20 mM imidazole). Subsequent gel filtration was carried out using an AKTA prime plus system with a Superose 6 column into a final buffer consisting of 20 mM Tris-HCl pH 8.0, 1 mM PMSF. For storage, sodium azide in a quantity of up to 3 mM and glycerol in a percentage of up to 40% were added to fractions before they were frozen with liquid nitrogen and stored at −80 °C.

### 4.7. MALDI-TOF Mass Spectrometry

Purified BlihIA complex was analyzed on 11.5% SDS-PAGE. The identity of protein bands was determined by matrix-assisted laser desorption/ionization time-of-flight (MALDI-TOF) mass spectrometry. Samples were prepared with Trypsin Gold (Promega, Madison, WI, USA) in accordance with the manufacturer’s instructions. Mass spectra were obtained using the rapifleX system (Bruker, Berlin, Germany).

## Figures and Tables

**Figure 1 ijms-26-08674-f001:**
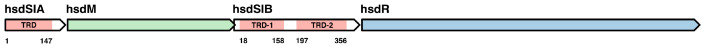
Organization of the BlihIA system. Amino acid positions of the TRD domains within HsdS subunits are indicated.

**Figure 2 ijms-26-08674-f002:**
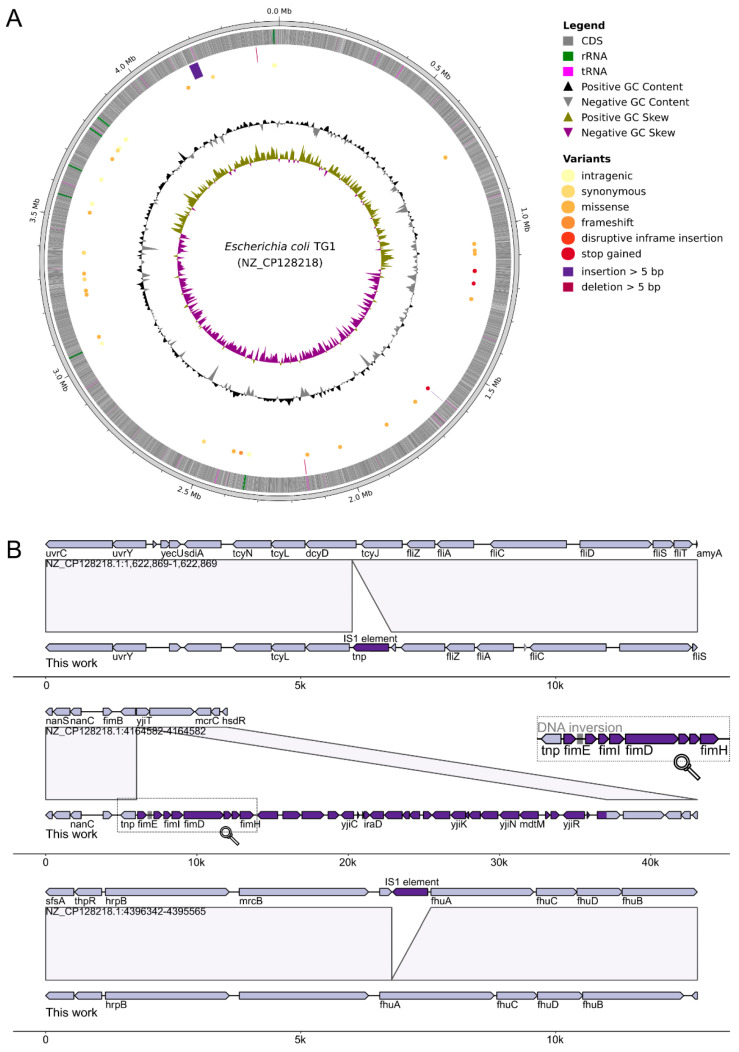
Alignment of TG1 genome assembled in this work vs. a reference TG1 genome. (**A**) Circos plot of *E. coli* strain TG1 genome (NZ_CP128218.1, F-plasmid excluded for clarity) with marked differences compared to in-house assembly. (**B**) Comparisons of large insertions and deletion positions between two assemblies. DNA inversion predicted with PhaVa v.0.2.3 [21].

**Figure 3 ijms-26-08674-f003:**
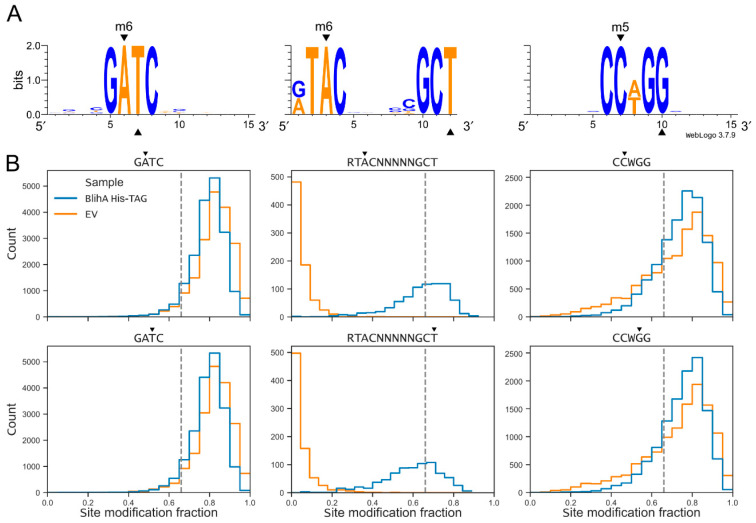
Determination of methylation motifs in genomic DNA of *E. coli* TG1 expressing BlihIA system or carrying an empty vector. (**A**) Methylation motifs predicted based on ONT sequencing data via MicrobeMod v.1.0.3 [23] and visualized with WebLogo v.3.7.9 [24] for cells with BlichIA system. (**B**) Distribution of site modification fraction of DNA from cells with and without RMI system. The site modification fraction is a fraction of reads with a detected methylation signal for one particular site within a genome. Black triangles indicate DNA modification positions.

**Figure 4 ijms-26-08674-f004:**
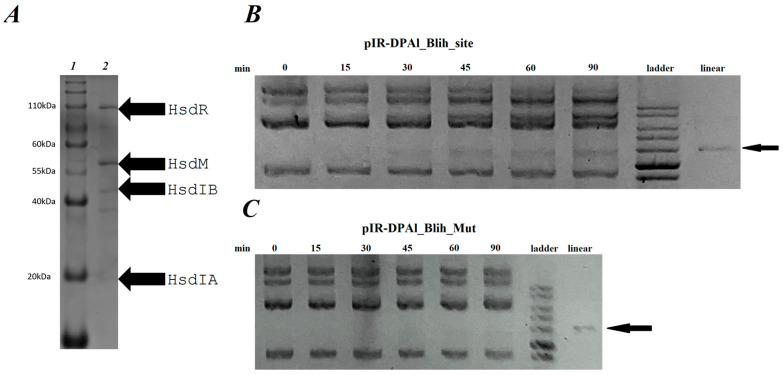
BlihIA in vitro. (**A**) SDS PAGE of BlihIA complex purified with Ni-NTA agarose and Superose 6. Lane 1—protein ladder 10–200 kDa, Servicebio (Wuhan, China); lane 2—purified BlihIA complex. (**B**) Agarose gel electrophoresis of the target plasmid restriction by BlihIA. Ladder—1kb DNA ladder (Evrogen, Moscow, Russia). Linear—linearized with HindIII pIR-DPAl_Blih_site, marked with an arrow. (**C**) Agarose gel electrophoresis. Enzymatic restriction of pIR-DPAl_Blih_Mut plasmid by BlihIA. Ladder—1kb DNA ladder (Evrogen, Moscow, Russia). Linear—linearized with HindIII pIR-DPAl_Blih_Mut, marked with an arrow.

**Figure 5 ijms-26-08674-f005:**
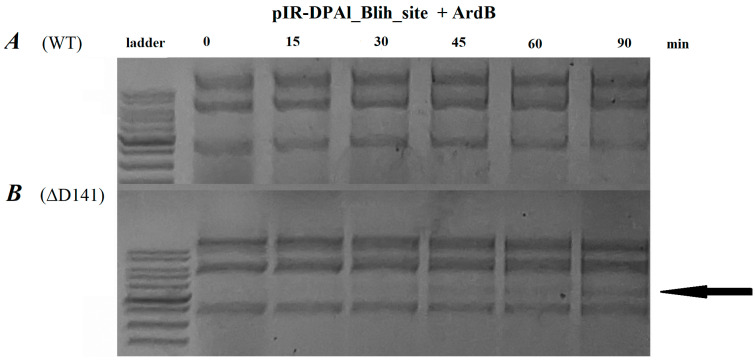
Agarose gel electrophoresis of the target plasmid restriction by BlihIA in the presence of ArdB (R64) (**A**) or its inactive form ArdBΔD141; (**B**) the linear form of pIR-DPAl_Blih_site (marked with an arrow).

**Table 1 ijms-26-08674-t001:** Lambda phage plaquing efficiency on *Escherichia coli* strains.

Strain	EOP *	*t*-Test
*E. coli* TG1	1	
*E. coli* TG1 pIRal-2_RM-Ia [18]	0.011 ± 0.007	The means are not significantly different at *p* < 0.05
*E. coli* TG1pIR-DPAl-BlihIA-His-TAG	0.022 ± 0.005	

* The results of five independent experiments are presented.

**Table 2 ijms-26-08674-t002:** Plasmids used in the present study.

Name	Resistance	Description
pIRal-2_RM-Ia [18]	Kn	Source of *blihIA* genes
pIR-DPAl [39]	Kn	Temperature-switchable acyl homoserine lactone-regulated expression vector
pIR-DPAl_Blich_site (this work)	Kn	pIR-DPAI based plasmid contains the single BlihIA recognition site
pIR-DPAl_Blich_site_Mut (this work)	Kn	pIR-DPAI-based plasmid contains the mutation in a single BlihIA recognition site

## Data Availability

ONT sequencing data including basecalled methylation signals and genome nucleotide sequences are available under BioProject: PRJNA1297036.

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
