# Peer review of "BlihIA—A Novel Type I Restriction-Modification System from Bacillus licheniformis Is Sensitive to In Vitro Inhibition by ArdB Antirestriction Protein"

_ijms, 2025, doi:10.3390/ijms26178674_

Round 1

Reviewer 1 Report

Comments and Suggestions for Authors

The functionality of the RMI system from B. licheniformis and the inhibitory action of the protein ArdB, as presented by the authors, are depicted with remarkable precision. This precision is expected to significantly contribute to a better understanding and practical application for transformation of the bacterial strain's functional characteristics. Furthermore, the conciseness of the overall explanation and the clear articulation of the findings are highly commendable aspects of the manuscript's organization.

Nonetheless, the terminology used in the manuscript is notably inconsistent or incorrectly presented in several instances, requiring proper revision. Furthermore, a significant omission is observed. It is also considerably regrettable that the experimentally measurement of improved transformation frequency utilizing this system, despite being stated as a research objective, was not performed. Therefore, it is anticipated that supplementing this result would substantially enhance the quality of the manuscript.

In the Discussion section, it is suggested that a comparative analysis with examples of systems possessing similar functionalities would likely offer a more comprehensive understanding of the genetic characteristics observed in the subject strain of this study.

In addition, please refer to the attached minor comments for further revisions. Specifically, it is imperative that the references are consistently formatted according to the journal's guidelines.

Major comments

  • Could be measure transformation frequency using BlihIA system and its inhibition by ArdB in licheniformis.
  • coli TG1 genome sequencing data analysis for BlihIA methylation site comparing between TG1 and BlihIA fully expressed. > then found IS1 element and a large 31-kb insert in fimB gene. Is this IS1 element translocated by cut-and-paste method?
  • In protein purification using BlihIA complex, Dps protein also co-purified. Why? Both proteins interact together? Are there some evidences? Need to discuss.
  • In results section, “Endonuclease activity by Lambda phage restriction assay and in vitro BlihIA complex” data was missed. Is it same to EOP counting and linear fragment production? Need to detail explanation the connection between two data.
  • References should be followed the journal guideline. Present list was the mixed form.

Minor comments

  • Line 23: Please ensure that the keywords are more comprehensively chosen, and a list of all abbreviations used in the manuscript should also be provided.
  • Lines 27, 30, 44, 47, 49, 53, 168: reference marking > need to change journal guideline ex. [1], [2] change to [1,2]
  • Line 35: kB should be change to “kilobases (kb)”
  • Line 60, 82, 89, 98: “TRD” already include domain, “EOP” assay, coli, “ONT” – The first instance of an abbreviation in the manuscript must be presented with its full term.
  • Line 61-62, 190-191: must be change the names of HsdSIAa and HsdS1Ab subunits (in text) to HsdSIA and HsdSIB (in figure 1), respectively. Or vis versa.
  • Line 79: hsdIA change to hsdSIA or hsdSIAb
  • Line 84: Please ensure that all cited references are properly indicated within the text, especially for pIRal-2_RM-Ia system
  • Line 97, 115: Type 1 R-M system or RMI system. Please ensure that a single, consistent nomenclature is used throughout.
  • Lines 142, 143: HsdIA > HsdSIA and HsdIB > HsdSIB
  • Line 146: Fig.4A > Fig. 4B
  • Line 158: “highly conserved C-terminal D141 residue” Is it point mutation or truncated mutation? If amino acid substitution, it is presented “D141x”. At line 159 and in figure 5B, ΔD141 indicates single amino acid deleted?
  • In Table 1, EOP values change to comma to dot. Also, indicate references
  • In Fig. 3 legend, BlichIA change to BlihIA in text. RM maybe change to RMI system
  • In Fig. 4A. What information is available regarding the unlabeled bands in Fig. 4A? Lane 1 and 2 missed. An explanation is needed concerning what the changing band patterns within the areas indicated by red circles signify in Fig, 4B.

  • In Table 2, any redundant content must be removed. In addition, plasmid vector for purification of BlihIA complex include in Table 2, for example, pIR-DPAl-BlihIA-His-TAG.
  • It is necessary to explicitly explain the significance of the supplementary files provided. Additionally, the Table descriptions should be clearly organized and presented.

Comments on the Quality of English Language

The terminology used in the manuscript is notably inconsistent or incorrectly presented in several instances, requiring proper revision. 

Author Response

Thank you very much for your comments.

Reviewer 2 Report

Comments and Suggestions for Authors

The article describes the discovery and functional characterization of a new restriction-modification system in the collection strain Bacillus licheniformis DSM13. Its in vitro inhibition by the antirestriction protein ArdB has also been demonstrated. The work contributes to a better understanding of the action of the RM I system in Gram-positive bacteria and presents several novel findings.
The obtained data support the main conclusions, but the results have significant shortcomings in their presentation.
I have the following comments and remarks:
1. The introduction is too short. There is no goal of the work; no goal/purpose/aim is formulated at all. There is no consistency in the presentation; everything is written briefly, as if the authors want to convey the information with a minimum amount of words, but this prevents understanding the meaning. Line 54: the authors say "we have cloned two novel RMI systems from B. licheniformis" - what did you clone into - species, strain, explain in a little more detail what you have obtained so far, and what are the novelties compared to your previous article [17].
2. The introduction lacks diagrams that would explain the meaning of the restriction-modification system. Give a diagram of the different systems - type I, type II, type III. Give similarities and differences between Gram-positive and Gram-negative bacteria. Describe your model system - E. coli, and explain why you use the lambda phage. This will make the article understandable to a broader audience.
3. The discussion is also very short. It should be expanded significantly for the article to be suitable for publication. Please refer to other articles published in IJMS.
Please provide parallels with evidence for blocking other RM systems by similar proteins. You must discuss the phenomenon of blocking BlihIA in vitro activity by ArdB protein and the lack of such in the absence of C-terminal D141. Why is this amino acid so essential? Well, show it on a 3D picture, for example, by 3D Swiss modeling.
4. The supplementary file is below the journal level. In this form, it is not possible to understand what the authors want to show. It should be formatted so that the reader can interpret the data. What do we see from these lines?
5. The methods are poorly described. The use of ONT sequencing, which the authors mention so many times, is everywhere as an abbreviation, and nowhere is it explained what the advantages of the method are (e.g., for identifying methylation sites, for mapping motifs, etc.). Please add the city and country of the companies.
6. When proposing strategies for circumventing RM barriers in Bacillus transformation, why not give a little more extensive applications of the genus in the discussion? In which biotechnologies are recombinant bacilli used? How exactly will your system support the process? The authors suggest the possibility of increasing the efficiency of transformation in Bacillus spp., but such attempts have not been made. Therefore, potential limitations or species-specific problems need to be discussed.
7. Although the co-purification of full-length and truncated HsdS subunits is intriguing, the manuscript would benefit from further discussion or hypotheses regarding the functional role of the partial HsdSIA. Is it modulatory, competitive, or involved in specificity switching? Even speculative models could guide future research.
8. Statistical processing: EOP assays and inhibitory assays need a clearer description of the statistical methods used, confidence intervals, or significance tests.

9. All references should be changed to the journal's format.

Comments on the Quality of English Language

The English needs to be significantly improved before the article can be accepted. There are meaningless words, fused words, and quite a few syntax and grammatical errors.

Author Response

Thank you very much for your comments.

Round 2

Reviewer 1 Report

Comments and Suggestions for Authors

I confirm that this manuscript has been thoroughly revised and significantly improved by incorporating the reviewers' suggestions. I anticipate that the publication of this paper will greatly contribute to academic advancement, particularly if it elucidates the operating principles of new RMI systems of Bacillus in the future. However, the citation formatting in the reference list and within the main text still causes some confusion, and I kindly request that the publishing office rectify these issues.

Author Response

Reviewer 1

I confirm that this manuscript has been thoroughly revised and significantly improved by incorporating the reviewers' suggestions. I anticipate that the publication of this paper will greatly contribute to academic advancement, particularly if it elucidates the operating principles of new RMI systems of Bacillus in the future. However, the citation formatting in the reference list and within the main text still causes some confusion, and I kindly request that the publishing office rectify these issues.

Thank you very much for your careful review. We have addressed the issues with the references.

Reviewer 2 Report

Comments and Suggestions for Authors

The authors have corrected many of the shortcomings of their work.

However, the introduction is not what I expected to see. The text "Here, we identified BlihIA system recognition site and confirmed it’s endonuclease activity in vitro. BlihIA immunity was previously shown to be sensitive to inhibition by ArdB antirestriction protein from the R64 conjugative plasmid 20, and here we confirmed that ArdB inhibits BlihIA DNA cleavage in vitro, thus demonstrating the in vitro antirestriction activity of the proteins from the ArdB/KlcA family for the first time."

should be changed to the following:

"The aim of the present work is to identify and confirm the BlihIA recognition site and to confirm whether ArdB inhibits BlihIA DNA cleavage in vitro."

Comments on the Quality of English Language

I continue to insist on improving the English language of the article.

Author Response

Reviewer 2

The authors have corrected many of the shortcomings of their work.

However, the introduction is not what I expected to see. The text "Here, we identified BlihIA system recognition site and confirmed it’s endonuclease activity in vitro. BlihIA immunity was previously shown to be sensitive to inhibition by ArdB antirestriction protein from the R64 conjugative plasmid 20, and here we confirmed that ArdB inhibits BlihIA DNA cleavage in vitro, thus demonstrating the in vitro antirestriction activity of the proteins from the ArdB/KlcA family for the first time."

should be changed to the following:

"The aim of the present work is to identify and confirm the BlihIA recognition site and to confirm whether ArdB inhibits BlihIA DNA cleavage in vitro."

Thank you very much for your thorough review. We changed the text according to your comment.

I continue to insist on improving the English language of the article.

The paper has been checked by a native English speaker. Grammar and stylistic mistakes have been corrected.